# Genome-wide meta-analysis implicates mediators of hair follicle development and morphogenesis in risk for severe acne

Christos Petridis[1], Alexander A. Navarini[1,2], Nick Dand [1], Jake Saklatvala[1], David Baudry[3], Michael Duckworth[3], Michael H. Allen[3], Charles J. Curtis[4,5], Sang Hyuck Lee[4,5], A. David Burden[6], Alison Layton[7], Veronique Bataille[8], Andrew E. Pink[3], The Acne Genetic Study Group[#], Isabelle Carlavan[9], Johannes J. Voegel[9], Timothy D. Spector[8], Richard C. Trembath [1], John A. McGrath [3], Catherine H. Smith[3], Jonathan N. Barker[3] & Michael A. Simpson[1]

Acne vulgaris is a highly heritable common, chronic inflammatory disease of the skin for which five genetic risk loci have so far been identified. Here, we perform a genome-wide association study of 3823 cases and 16,144 controls followed by meta-analysis with summary statistics from a previous study, with a total sample size of 26,722. We identify 20 independent association signals at 15 risk loci, 12 of which have not been previously implicated in the disease. Likely causal variants disrupt the coding region of *WNT10A* and a P63 transcription factor binding site in *SEMA4B*. Risk alleles at the 1q25 locus are associated with increased expression of *LAMC2*, in which biallelic loss-of-function mutations cause the blistering skin disease epidermolysis bullosa. These findings indicate that variation affecting the structure and maintenance of the skin, in particular the pilosebaceous unit, is a critical aspect of the genetic predisposition to severe acne.

[1] Department of Medical and Molecular Genetics, School of Basic & Medical Biosciences, King's College London, London SE1 9RT, UK. [2] Departement of Dermatology, University Hospital of Zurich and University of Zurich, CH-8091 Zurich, Switzerland. [3] St John's Institute of Dermatology, School of Basic & Medical Biosciences, King's College London, London SE1 9RT, UK. [4] NIHR Maudsley Biomedical Research Centre (BRC) at South London and Maudsley NHS Foundation Trust (SLaM) & Institute of Psychiatry, Psychology and Neuroscience (IoPPN), King's College London, London SE5 8AF, UK. [5] Social Genetic & Developmental Psychiatry Centre, Institute of Psychiatry, Psychology and Neuroscience (IoPPN), King's College London, London SE5 8AF, UK. [6] Institute of Infection, Immunity and Inflammation, University of Glasgow, Glasgow G12 8TA, UK. [7] Department of Dermatology, Harrogate and District Foundation Trust, Harrogate HG2 7SX, UK. [8] Twin Research and Genetic Epidemiology Unit, School of Basic & Medical Biosciences, King's College London, London SE1 7EH, UK. [9] Research Department, Galderma R&D, Sophia Antipolis 06410 Biot, France. A full list of consortium members appears at the end of the paper. Correspondence and requests for materials should be addressed to J.N.B. (email: jonathan.barker@kcl.ac.uk) or to M.A.S. (email: michael.simpson@kcl.ac.uk)

Acne vulgaris is an inflammatory disease of the skin, primarily affecting the face, chest and back. The biological mechanisms that lead to lesion development are poorly understood, but involve a complex interplay between sebum production, follicular keratinisation, inflammation, and colonisation of pilosebaceous follicles by *Propionibacterium acnes*[1]. The characteristic inflammatory papules, pustules and nodules typically first develop during puberty, may persist for decades and leave disfiguring scars in up to 20% of patients. Acne can have severe emotional and psychological consequences and has been associated with depression, unemployment, suicidal ideation and suicide itself[1]. Severe acne is typically treated with topical and systemic agents that suppress the microbiome repertoire or the activity of sebaceous glands. The treatment regimes are often ineffective and poorly tolerated, and there remains a substantial unmet medical need.

Evidence of a genetic component to acne susceptibility is well established[2] and previous genome-wide association studies (GWAS) of severe acne have identified three genomic loci harbouring alleles that are associated with the disease in the European population[3], and two in the Han Chinese population[4]. These loci have provided insight into the biological mechanisms that underlie disease pathogenesis, including a potential role for components of the TGFβ pathway.

In the current study, we further delineate the genetic susceptibility of severe acne through the identification of genetic variation at 15 genomic loci that contribute to disease risk. Investigation of the consequence of the associated alleles at these loci indicates that the contribution to acne susceptibility may be, at least in part, mediated through variation in the structure and maintenance of the pilosebaceous unit in the skin.

## Results

**Genome-wide association study and meta-analysis**. To investigate the genetic basis of acne we have performed a GWAS of 3823 severe acne cases, recruited through a network of hospital-based dermatologists within the United Kingdom, and 16,144 unselected population controls (Supplementary Table 1). Following quality control and genome-wide imputation we tested more than 7.4 million SNPs for association with acne. At the three loci (1q41, 5q11.2 and 11q13.1) harbouring acne-associated alleles in an independent UK acne study population[3], we observed strong evidence of association with a consistent direction and magnitude of effect as was previously reported (Table 1, Supplementary Table 2). However, we did not replicate the associations at 1q24.2 or 11p11.2 described in the Han Chinese population[4], highlighting potential trans-ethnic differences in the genetic contributors to acne susceptibility (Supplementary Table 2).

We undertook a meta-analysis using summary statistics from this newly performed GWAS and the previously published GWAS of severe acne in the UK population[3], yielding a combined sample size of 5602 severe acne cases and 21,120 population controls (Methods, Supplementary Figure 1). We observed moderate inflation of test statistics ($\lambda_{GC} = 1.09$, Supplementary Figure 2) but LD score regression indicated that this inflation is driven by trait polygenicity rather than confounding bias (LD score regression intercept = 1.025). We observed genome-wide significant association with acne susceptibility at 15 independent genomic loci, of which 12 have not been reported previously (Table 1, Supplementary Figure 1). The magnitude and direction of effect of the lead variant at each of the observed risk loci are consistent between the two studies (Supplementary Figure 3). To determine the presence of statistically independent associations with disease risk at each of these loci, we undertook a series of stepwise conditional analyses. Evidence for a second conditionally independent association signal was observed at three loci (2q35, 11q13.1, and 15q26.1) with evidence for three distinct SNP association signals at 1q41 (Supplementary Table 3), giving a total of 20 independent acne associations across the 15 loci. There is no evidence of epistasis between the associated loci.

**Investigation of associated loci**. We noted that one of the newly identified acne susceptibility loci is located within the common ~ 3.8 Mb[5] inversion at 8p23.1 (rs28570522, OR = 1.14, 95% CI 1.10–1.20, $P = 1.31 \times 10^{-9}$, Table 1). The inversion region displays extended linkage disequilibrium (LD), driven by suppression of local recombination between the non-collinear regions in

---

### Table 1 Variants with the strongest evidence of association in each of the 15 acne-associated loci

| SNP ID | Chr | Position (hg19) | Band | RA | PA | RAF cases | RAF controls | Navarini P | Navarini OR (95% CI) | New GWAS P | New GWAS OR (95% CI) | Meta P | Meta OR (95% CI) | Implicated gene |
|---|---|---|---|---|---|---|---|---|---|---|---|---|---|---|
| rs10911268 | 1 | 183,122,718 | 1q25.3 | C | A | 0.63 | 0.60 | 0.002661 | 1.13 (1.04-1.23) | $2.44 \times 10^{-10}$ | 1.19 (1.13-1.25) | $3.88 \times 10^{-12}$ | 1.17 (1.12-1.22) | *LAMC2* |
| rs788790 | 1 | 202,289,606 | 1q32.1 | C | A | 0.53 | 0.50 | $7.41 \times 10^{-5}$ | 1.17 (1.08-1.26) | $1.96 \times 10^{-5}$ | 1.12 (1.06-1.17) | $9.39 \times 10^{-9}$ | 1.13 (1.09-1.18) | *LGR6* |
| rs1256580 | 1 | 219,199,380 | 1q41[a] | C | G | 0.18 | 0.15 | 0.00222 | 1.17 (1.06-1.30) | $1.12 \times 10^{-9}$ | 1.23 (1.15-1.31) | $1.23 \times 10^{-11}$ | 1.21 (1.15-1.28) | *TGFB2* |
| rs2901000 | 2 | 60,501,216 | 2p16.1 | A | G | 0.46 | 0.43 | $1.04 \times 10^{-5}$ | 1.19 (1.10-1.29) | $2.53 \times 10^{-8}$ | 1.16 (1.10-1.22) | $1.50 \times 10^{-12}$ | 1.17 (1.12-1.22) | *BCL11A*[b] |
| rs1092479 | 2 | 121,769,437 | 2q14.2 | C | G | 0.30 | 0.27 | 0.000224 | 1.17 (1.08-1.28) | $3.60 \times 10^{-5}$ | 1.12 (1.06-1.19) | $4.30 \times 10^{-8}$ | 1.14 (1.09-1.19) | *GLI2*[b] |
| rs121908120 | 2 | 219,755,011 | 2q35 | T | A | 0.98 | 0.97 | 0.001637 | 1.66 (1.21-2.27) | $1.40 \times 10^{-10}$ | 2.10 (1.67-2.63) | $1.82 \times 10^{-12}$ | 1.94 (1.61-2.33) | *WNT10A* |
| rs4487353 | 4 | 124,253,789 | 4q27-28.1 | G | A | 0.36 | 0.33 | 0.000115 | 1.17 (1.08-1.27) | $3.83 \times 10^{-6}$ | 1.13 (1.07-1.19) | $2.32 \times 10^{-9}$ | 1.14 (1.09-1.20) | *FGF2* |
| rs629725 | 5 | 52,631,067 | 5q11.2[a] | T | C | 0.37 | 0.33 | $4.69 \times 10^{-5}$ | 1.18 (1.09-1.28) | $8.22 \times 10^{-12}$ | 1.20 (1.14-1.27) | $1.90 \times 10^{-15}$ | 1.20 (1.14-1.25) | *FST* |
| rs158639 | 5 | 55,611,710 | 5q11.2 | A | G | 0.30 | 0.27 | 0.007973 | 1.12 (1.03-1.22) | $9.22 \times 10^{-7}$ | 1.15 (1.09-1.21) | $2.70 \times 10^{-8}$ | 1.14 (1.09-1.19) | |
| rs7809981 | 7 | 40,874,376 | 7p14.1 | T | G | 0.26 | 0.24 | 0.00213 | 1.15 (1.05-1.26) | $3.74 \times 10^{-6}$ | 1.15 (1.08-1.22) | $2.82 \times 10^{-8}$ | 1.15 (1.09-1.20) | |
| rs28570522 | 8 | 10,630,568 | 8p23.1 | A | G | 0.40 | 0.37 | 0.000821 | 1.14 (1.06-1.24) | $4.206 \times 10^{-7}$ | 1.14 (1.09-1.21) | $1.31 \times 10^{-9}$ | 1.14 (1.10-1.20) | |
| rs2727365 | 11 | 13,111,484 | 11p15.3-15.2 | G | A | 0.36 | 0.32 | 0.000426 | 1.16 (1.07-1.25) | $1.08 \times 10^{-10}$ | 1.19 (1.13-1.25) | $2.28 \times 10^{-13}$ | 1.18 (1.13-1.23) | |
| rs144991069 | 11 | 64,827,708 | 11q13.1-13.2[a] | A | T | 0.02 | 0.01 | $8.91 \times 10^{-6}$ | 2.08 (1.50-2.86) | $1.01 \times 10^{-8}$ | 1.85 (1.50-2.28) | $5.00 \times 10^{-13}$ | 1.91 (1.60-2.28) | *OVOL1, MAPK11* |
| rs34560261 | 15 | 90,734,426 | 15q26.1 | C | T | 0.86 | 0.83 | 0.000445 | 1.25 (1.10-1.41) | $1.82 \times 10^{-12}$ | 1.35 (1.24-1.47) | $5.89 \times 10^{-15}$ | 1.32 (1.23-1.41) | *SEMA4B* |
| rs28360612 | 22 | 24,883,218 | 22q11.23 | T | A | 0.74 | 0.72 | 0.004014 | 1.14 (1.04-1.25) | $7.12 \times 10^{-7}$ | 1.16 (1.09-1.23) | $1.05 \times 10^{-8}$ | 1.15 (1.10-1.21) | *SPECC1L* |

Chr: chromosome, RA: risk allele, PA: protective allele, RAF: risk allele frequency, OR: odds ratio, 95% CI: 95% confidence interval
[a]Previously reported acne susceptibility locus (Navarini et al.)
[b]Implicated through relationship to the sparse hair (MP:0000416) gene-set by DEPICT

inversion heterozygotes. This extended LD is reflected in the observed pattern of acne association in the region, with strong evidence of association observed across multiple SNPs spanning the entire inversion region (Supplementary Figure 4). To further investigate the allelic nature of acne susceptibility at the 8p23.1 inversion, we inferred inversion genotypes in our study population. Association analysis of the inversion status indicated that the derived non-inverted haplotype (orientation consistent with the reference genome) is associated with increased acne risk (OR = 1.11, 95% CI 1.06–1.16, $P_{meta} = 2 \times 10^{-6}$) and conditioning on the inversion genotype reduces the strength of the observed SNP associations (rs28570522, OR = 1.12, 95% CI 1.06–1.19, $P_{conditional} = 0.00011$). Taken together these data suggest that the causal acne risk allele at this locus resides more commonly on the derived non-inverted background. This orientation of the inversion haplotype has also been previously reported to be associated with susceptibility to SLE[6] and rheumatoid arthritis[7], but a protective effect has recently been observed in a GWAS of neuroticism[8]. It is also of potential relevance that the inversion locus harbours the copy number variable β-defensin gene cluster. The transcriptional activity of this cluster has been previously demonstrated to be upregulated in acne lesions[9].

To highlight putative causal variants at each of the 14 other susceptibility loci, we performed Bayesian summary statistic fine-mapping to identify credible sets of variants likely to underlie the observed association signals (Methods and Supplementary Table 4). Two association signals mapped to single variants with > 50% posterior probability of being causal: rs121908120 ($P_{causal} = 0.88$), a missense variant in WNT10A (Fig. 1 and Supplementary Table 4), and rs34560261 ($P_{causal} = 0.66$) located in intron 1 of SEMA4B (Fig. 2 and Supplementary Table 4). The missense allele (p.F228I) in WNT10A has a frequency of 0.03 in the control population and exerts a protective effect on acne (Table 1). WNT10A encodes a member of the Wnt family of secreted signalling proteins that contribute to the regulation of cell fate and patterning[10]. Notably, Wnt-10a itself is strongly expressed in the dermal papilla within the pilosebaceous unit during the anagen phase of hair growth and is expressed in the dermal condensate and the adjacent follicular epithelium[11]. The p.F228I missense allele was originally identified as the most frequently observed causal allele in a recessive form of ectodermal dysplasia (OMIM: 257980) characterised by abnormal development of ectodermal derivatives including hair, teeth, sweat glands and nails[12]. Refinement of the phenotypic effects of the p.F228I allele in this clinical context revealed that both dry skin and sparse hair are recurrently observed in homozygous individuals, but also often in heterozygous carriers[13]. The reduced hair follicle activity and sebum production that results from a disruption of Wnt-10a activity is therefore consistent with the observed protective effect of the p.F228I allele on acne development. The recessive ectodermal dysplasia caused by biallelic disruption of WNT10A is typically both more severe and widespread in males than in females[13]. In our acne study population the p.F228I allele has a strong effect in both males and females (OR$_{males}$ = 2.86, 95% CI = 2.08–3.94, $P = 1.12 \times 10^{-10}$ and OR$_{females}$ = 1.53, 95% CI 1.22–1.92, $P = 0.00027$), but notably the observed effect is significantly larger in males than females ($P = 0.0018$, Fig. 1). We also note a comparable sex bias in effect size at the other conditionally independent acne association at this locus (rs72966077; OR$_{males}$ = 1.56, 95% CI = 1.31–1.86, $P = 6.02 \times 10^{-7}$ and OR$_{females}$ = 1.18, 95% CI 1.02–1.36, $P = 0.024$). However, we do not observe a sex bias in observed effect sizes at any of the other 14 acne associated loci (Supplementary Figure 5), nor do we identify any additional sex-specific genome wide significant acne signals.

The second putative causal variant identified by the fine-mapping approach is rs34560261, located in intron 1 of SEMA4B at 15q26.1. Whilst little is known about the biological function of SEMA4B itself, the variant is located within a site at which the TP63 transcription factor has been demonstrated to bind in keratinocytes[14] within a broader region of DNAase hypersensitivity (Fig. 2). The binding site harbours a conserved sequence motif that is disrupted by rs34560261 (Fig. 2). The protective minor allele introduces a thymine nucleotide at the position of an invariant cytosine (Fig. 2), which is predicted to ablate the TP63-binding potential of this sequence (Fig. 2, risk allele sum occupancy score = 6287.91, protective allele sum occupancy score = 2.18). The transcription factor TP63 is critically important for epidermal morphogenesis including hair follicle development[15] and rare mutations in the TP63 gene have also been described in monogenic ectodermal dysplasia syndromes that have substantial phenotypic overlap with ectodermal dysplasias resulting from mutation of WNT10A[16]. There is strong evidence that the acne association signal at this locus and a skin eQTL for SEMA4B colocalise ($P_{coloc} = 0.98$), with the allele that ablates the TP63-binding motif associated with a reduction in SEMA4B expression in skin and conferring protection against severe acne.

Across the remaining 12 loci, statistical fine-mapping did not clearly resolve the association signals to individual causal variants. However, through regional colocalisation with skin eQTLs (Methods and Supplementary Table 5) we were able to identify putative causal genes at additional acne risk loci including a series of genes with established roles in skin biology and pathology. An eQTL for LAMC2 in skin colocalises with the acne association signal at 1q25.3 ($P_{coloc} = 0.97$). LAMC2 encodes a component of the extracellular matrix glycoprotein Laminin-5 that is strongly expressed in the epithelia of all tissues[17]. Biallelic loss-of-function alleles in LAMC2 are an established cause of generalised severe junctional epidermolysis bullosa (OMIM: 226700), an extreme form of inherited skin and mucous membrane fragility and blistering that is associated with a reduced life expectancy[18]. In contrast, at this locus the acne risk haplotype is associated with increased expression of LAMC2 in the skin, providing insight into the phenotypic consequence of the opposite extreme of an allelic series in this gene. Colocalisation further implicated genes with established roles in skin biology at several other newly identified acne susceptibility loci: LGR6 at 1q32.1, which encodes a glycoprotein hormone receptor that is observed to be strongly expressed by cells in the stem cell niche within the pilosebaceous unit in mice;[19] FGF2 at 4q28.1, which has established roles in wound healing and scarring, and SPECC1L at 22q11.23, which has previously been identified as the site of rare pathogenic mutations in forms of oblique facial clefting (OMIM: 600251). OVOL1 was previously suggested as a potential candidate gene at the acne susceptibility locus at 11q13.1–13.2. However, we note a strong colocalisation between the acne association and an eQTL for MAP3K11, which encodes a stress-responsive protein kinase; this offers an alternative potential biological mechanism through which acne susceptibility is mediated by variation at this locus.

**Implicated biological pathways.** The identification of this series of putative causal genes with established roles in skin and hair biology highlights the importance of pilosebaceous unit development and morphogenesis in the aetiology of acne. Taken in this context, the implication of genes within the TGFβ pathway at acne risk loci previously reported in the UK population potentially adds further support to the relevance of this biological process. The TGFβ pathway is involved in a range of biological processes across tissue and cell types and both TGFB2 (1q41) and

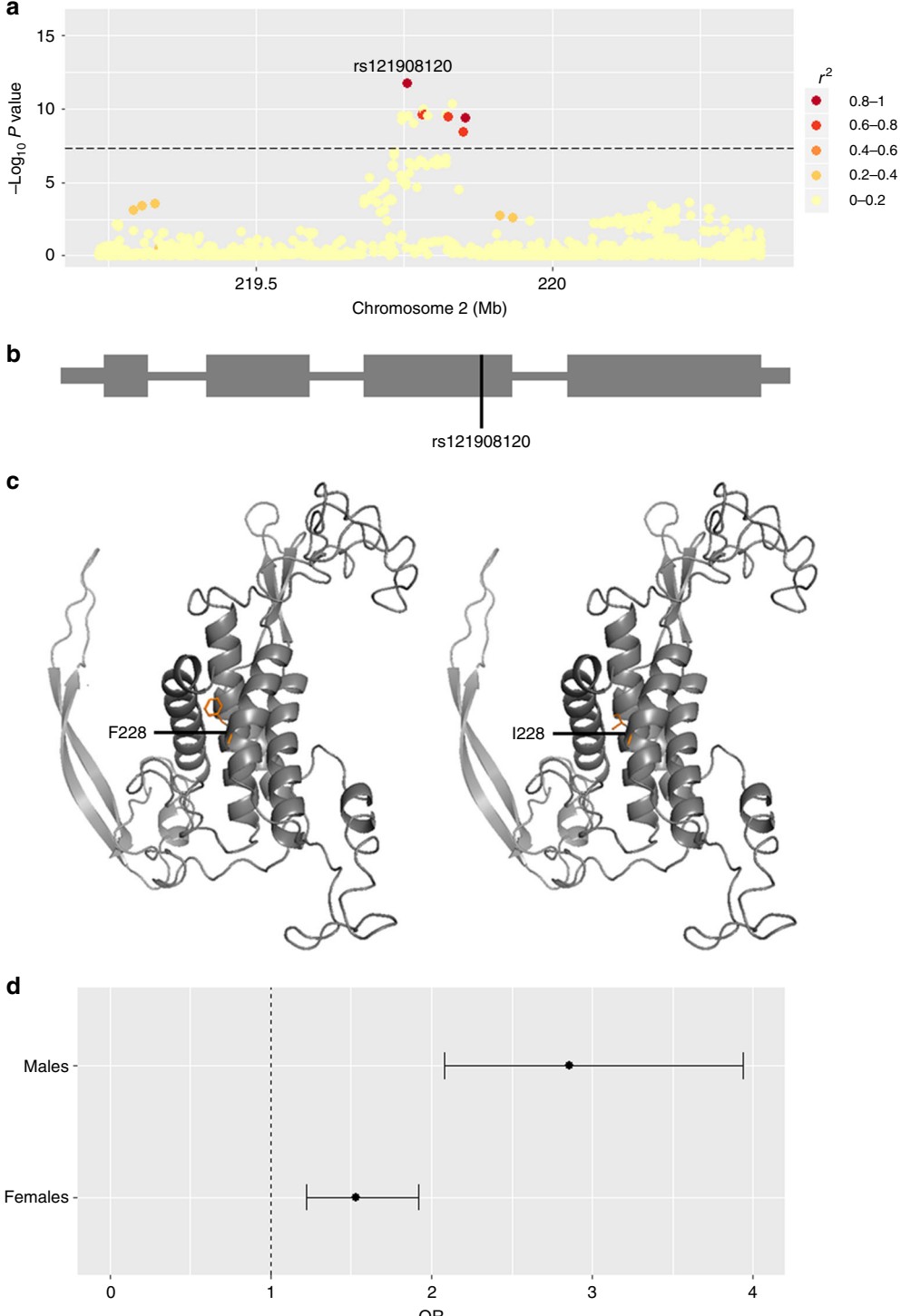

**Fig. 1** Acne association at the 2q35 locus (*WNT10A*). **a** Regional acne association plot, variants coloured according to the degree of LD (*r*[2]) with the lead SNP (rs121908120, *WNT10A*:p.F228I). **b** Location of the rs121908120, p.F228I with respect to the *WNT10A* gene structure. **c** Partial three dimensional predicted protein structure incorporating phenylalanine (risk, left) or isoleucine (protective, right) at position 228 (SWISS-MODEL repository). **d** Forest plot indicating the difference in the effect size of the acne association between males and females for rs121908120. Error bars represent 95% confidence intervals for estimated odds ratios

*FST* (5q11.2) have been identified as mediators of the morphological changes that occur through the hair follicle cycle[20,21].

To identify candidate causal genes in these or other related biological pathways at remaining acne susceptibility loci, we deployed a bioinformatics approach to establish whether an enrichment of genes with related biological function was observed

(Methods). The approach was applied to the 15 loci harbouring genome-wide significant associations and a further 54 loci at which allelic associations with acne meeting a less stringent threshold of statistical significance ($P < 1 \times 10^{-5}$) were observed (Supplementary Table 6). Enrichment (FDR < 0.05) of 15 gene-sets was observed, including gene-sets relating to branches of the

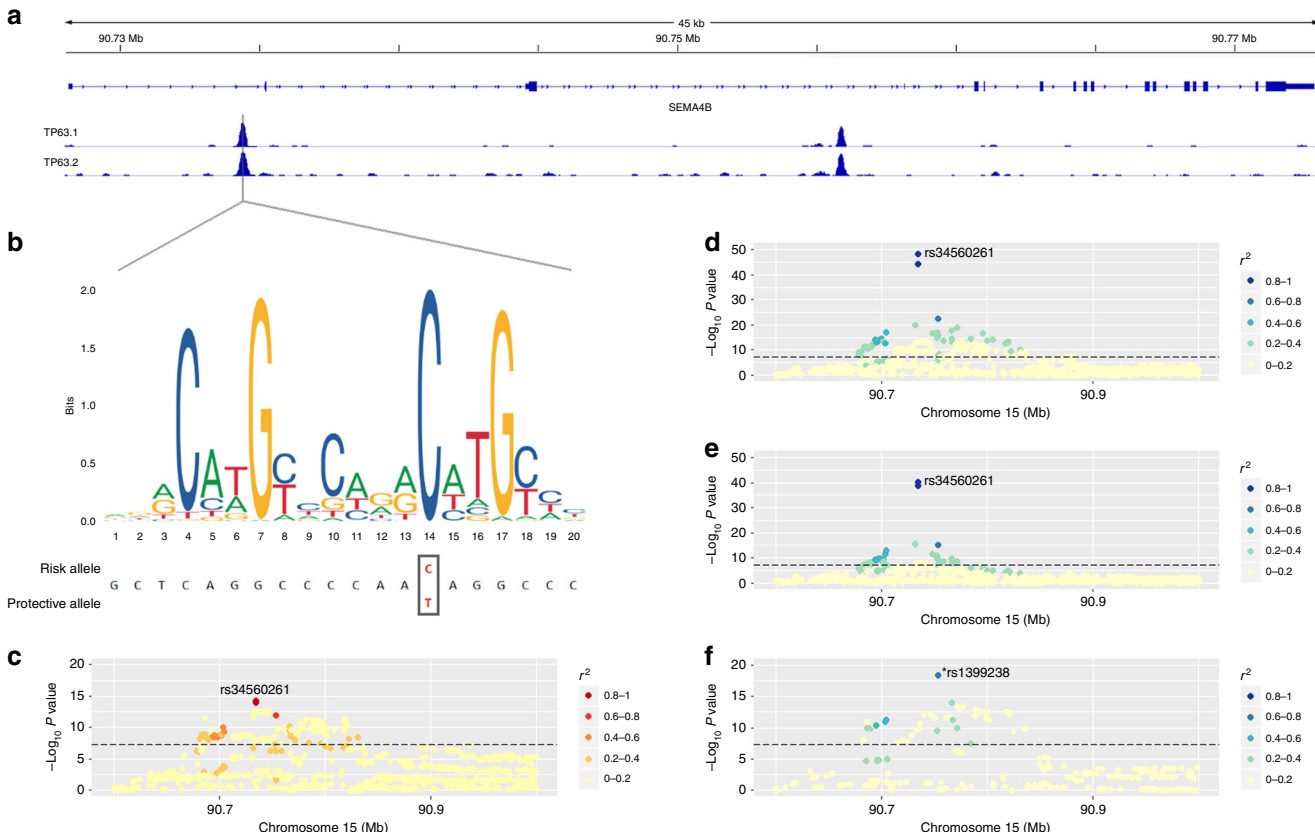

**Fig. 2** Acne association signal at the 15q26.1 locus (*SEMA4B*). **a** ChIP-Seq read signal intensities of TP63 binding on two samples from human neonatal foreskin keratinocytes. **b** Sequence logo of the *TP63*-binding motif highlighting the risk and protective alleles for rs34560261. The invariant cytosine (position 14) is substituted by a thymine. **c** Regional acne association plot, variants are coloured according to the degree of LD ($r^2$) with the lead SNP (rs34560261). **d** *SEMA4B* not sun exposed skin eQTL from GTEx. **e** *SEMA4B* sun exposed skin eQTL from GTEx. **f** *SEMA4B* skin eQTL from MuTHER study; * rs1399238 in LD with rs34560261 ($r^2 = 0.66$) (rs34560261 was not present in the MuTHER dataset)

mammalian phenotype ontology tree that describe abnormal epidermal and ectodermal development (Supplementary Table 6); these implicate candidate causal genes with related biological functions at otherwise unresolved acne risk loci, including *BCL11A* at 2p16.1 and *GLI2* at 2q14.2 (Table 1, Supplementary Table 6). Despite limited direct evidence implicating immune-related genes and pathways in acne susceptibility, estimation of genetic correlation of severe acne with 175 other traits (Methods) reveals evidence of genetic correlation with inflammatory bowel disease (Supplementary Data 1)[22] suggesting that there may be elements of shared genetic aetiology with this immune-mediated disease.

## Discussion

The current study provides a substantial advance in our insight into the genetic susceptibility and pathogenic mechanisms that contribute to the development of severe acne, increasing the number of genomic loci at which genetic variation is robustly associated with acne susceptibility in the European population from 3 to 15. Approximately 22% of the phenotypic variance is explained by variants across the genome that were examined in this study. The combination of the 15 genome-wide significant loci accounts for ~3% of the phenotypic variance, indicating that there are further loci contributing to the disease susceptibility that remain undiscovered. Fine-mapping and eQTL colocalisation of the identified association signals have enabled the implication of genes including *WNT10A*, *LGR6*, *TP63* and *LAMC2* that have established roles in controlling the development, morphology and activity of hair follicles. The identification of this series of putative

causal genes provides the basis for an appealing hypothesis that genetic susceptibility to acne results, in part, from variation in the structure and maintenance of the pilosebaceous unit that creates a follicular environment prone to bacterial colonisation and resulting inflammation. This insight highlights processes that contribute to hair follicle development and maintenance as potential therapeutic targets to complement current therapeutic regimes that focus on suppression of inflammation and bacterial colonisation.

## Methods

**Clinical resource**. The study was designed in accordance with the Declarations of Helsinki, and ethical approval was obtained from the NRES Committee London-Westminster (reference CLRN 05/Q0702/114). Individuals with severe acne were recruited through a network of 45 dermatology centres in the UK. Each participant provided signed consent and a clinical assessment was undertaken by a trained dermatologist. The diagnostic criteria were the same as previously employed in Navarini et al.[3], with one or more of the following criteria required for diagnosis: (a) nodulocystic disease; (b) ≥5 points in any body region assessed by the validated Leeds clinical acne score that uses a colour photographic acne grading scheme to evaluate the severity of involvement of body regions (face 0–12, chest 0–8 and back 0–8);[23] (c) requiring treatment with isotretinoin; and (d) presence of rare and severe forms of acne.

**Genotyping and quality control**. Genome wide genotyping of the case cohort was undertaken in two batches using the Illumina Human Omni Express Exome 8v1.2 (2567 cases) and Illumina Infinium Omni Express Exome 8v1.3 (1961 cases). Genotype calling was performed using the Genome Studio Software package (Illumina). Control genotypes were obtained from the English Longitudinal Study of Aging (ELSA, genotyped on the Illumina Human Omni 2.5) and the Under-standing Society Project (USP, genotyped on the Illumina Human Core Exome v12.0). Both control cohorts are unselected population control cohorts. Quality

control was performed in two batches, one containing genotypes from 2567 cases and 7452 controls from the ELSA and the second with 1961 cases and 9500 controls from the USP.

Variants were excluded if they were only genotyped in either cases or controls within each batch, had a call rate <0.99 or a significant difference in call rate between cases and controls ($P < 5 \times 10^{-7}$), or deviated from Hardy–Weinberg equilibrium ($P < 1 \times 10^{-4}$). Individuals with a call rate < 0.99 or heterozygosity estimates that deviate more than five standard deviations from the mean were excluded. Ancestry outliers were detected with principal component analysis (KING v1.4) and excluded from downstream analysis[24]. Genetic relatedness between individuals within the combined cohort and the cohort reported by Navarini et al.[3] was estimated and all but one individual from groups of related individuals (kinship coefficient > 0.0442, estimated > third degree relatives) were excluded[24]. Following quality control batch 1 comprised 358,871 successfully genotyped variants in 1996 cases and 6978 controls and batch 2 comprised 229,556 successfully genotyped variants in 1827 cases and 9166 controls.

**Imputation**. Phasing and imputation of the two study batches were undertaken using the Haplotype Reference Consortium (HRC version r1.1) reference panel on the Michigan Imputation Server[25]. Post imputation, variants with info score < 0.7 or a minor allele frequency (MAF) of < 0.005 in either study batch were excluded from downstream analysis, resulting in a combined total of 7,877,859 variants successfully genotyped or imputed in a combined total of 3823 cases and 16,144 controls.

**Association analysis**. Association testing was performed with a logistic Wald association test (EPACTS), including the first four principal components and QC/imputation batch as covariates.

**Meta-analysis**. Results from the association analysis were included in a standard error-weighted meta-analysis with GWAS summary statistics from a previous study of 1779 acne cases and 4976 controls in the UK population (Navarini et al.[3]), performed with METAL (release 2011-03-25)[26]. Variants with evidence of heterogeneity between the two studies ($P$-het < 0.05) or with a MAF < 0.005 were excluded from further analysis, resulting in a total of 7,441,713 variants utilised in downstream studies.

**LD score regression**. Linkage disequilibrium score regression was performed using LDSC v1.0.0 software using summary statistics on variants that had been directly genotyped or imputed with INFO > 0.95[27]. LD score regression was also used to estimate the genetic correlation between severe acne and 175 different phenotypes for which GWAS have been performed in European populations. This analysis was performed through the Ldhub interface (http://ldsc.broadinstitute.org/ldhub/)[28,29].

**Chromosome 8 inversion genotyping**. The orientation of the 3.8 Mb segment on chromosome 8 was inferred using a total of 736 variants that lie within the boundaries of the inversion. The posterior probability of each the three possible inversion genotypes (N/N, N/I and I/I; N = not inverted, I = inverted) for each individual was calculated from the first principal component calculated across these 736 variants (KING[24]) using a Gaussian mixture model fitted with an EM algorithm (R package mixtools[30]).

**Transcription factor motif analysis**. To evaluate the effect of the single variant substitution on *TP63*-binding capacity, the sum occupancy score[31] for both alleles was calculated with PWMtools[32] from the *TP63* nucleotide position weight matrix (PWM) from the JASPAR database[33].

**Locus definition**. An LD window was calculated for every variant with a meta-analysis $P$ value of < $5 \times 10^{-8}$, defined by the most proximal and distal variants with an $r^2$ of > 0.5. LD was calculated in the GBR and CEU samples from the 1000 Genomes Phase 3[34]. Regions were combined if there was < 500 Kb between neighbouring LD-defined regions. The variant with the strongest evidence of association was considered the lead variant for each locus.

**Conditional analysis**. Stepwise conditional analysis was performed at each associated locus (EPACTS). The genotypes of the variant with the strongest evidence of association were sequentially included as covariates in iterated logistic regression models. This process was performed at each locus in the newly generated dataset and Navarini et al. dataset and combined through a standard error-weighted meta-analysis (METAL). At each locus this process was repeated until there were no remaining variants that had evidence of association (meta-analysis $P < 5 \times 10^{-5}$).

**Fine-mapping**. An approximate Bayes factor was calculated from the effect size and standard error of each variant in each associated locus, using the approach defined by Wakefield:[35] $\mathrm{ABF} = \sqrt{\frac{V+W}{V}} \exp\left(-\frac{z^2}{2}\frac{W}{(V+W)}\right)$, assuming a prior variance on the log odds ratios of 0.04. The resulting Bayes factors were then rescaled to reflect the posterior probability for each variant being causal and 95% credible sets were defined as the minimal set of variants whose combined posterior probabilities sum to ≥0.95.

**eQTL colocalisation**. Estimation of the colocalisation between acne association signals and skin cis-eQTLs from the MuTHER Study[36] and GTEx[37] was performed. Candidate skin eQTLs were defined as any variant located within an acne risk locus that was also associated with variation in the expression of a nearby gene ($\pm 1$ Mb, $P < 1 \times 10^{-4}$). A Bayesian test for colocalisation between the acne association signal and the skin eQTL signal was performed using a set of variants that overlapped between the two studies using the R package coloc[38], with a prior probability of colocalisation defined as $P$: $10^{-5}$. In the MuTHER dataset, if multiple eQTL signals for the same gene had been generated using different gene expression assay probes, then the test for colocalisation was performed with each probe association signal separately.

**Biological pathway analysis**. DEPICT was used to undertake gene prioritisation for regions of genome-wide significance and to investigate over-representation of genes within biological pathways[39]. This method uses prior information to quantify evidence for membership of genes in predefined gene-sets including molecular pathways, tissue-specific expression gene-sets and gene-sets relating to specific biological processes. As recommended[39], two separate analyses were conducted, for loci with $P < 1 \times 10^{-5}$ and $P < 5 \times 10^{-8}$. 5000 permutations were conducted to adjust the enrichment $P$ values for biases and a further 500 permutations to define the false discovery rate.

## Data availability
Full meta-analysis summary statistics are available at the European Genome-phenome Archive under the collection ID EGAS00001003278.

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

## Acknowledgements

We acknowledge support from the National Institute for Health Research (NIHR), through the Dermatology Clinical Research Network; the NIHR Biomedical Research Centre based at Guy's and St Thomas' NHS Foundation Trust and King's College London; a research grant from Galderma SA to J.N.B., C.H.S. and M.A.S.; Bruno Bloch, Promedica Foundation and HSM-2 canton of Zurich grant to A.A.N.; the NIHR Maudsley Biomedical Research Centre at South London and Maudsley NHS Foundation Trust and King's College London; capital equipment funding from the Maudsley Charity (Grant Ref. 980) and Guy's and St Thomas' Charity (Grant Ref. STR130505). We thank Ulrike Blume-Peytavi; Leaca Crawford; Jana Estafan; Darren Geoghegan; Dan Glass; Alison Gosh; Naomi Hare; Helen Holmes; Karen Markwel; Philippe Martel; Carine Marty; Corinne Ménigot; Anne Thompson; Kate Thornberry; Bianca Tobin, and the participating patients and supporting staff in all the study centres.

## Author contributions

C.P., A.A.N., N.D., J.S., D.B., M.D., C.J.C., S.H.L., I.C., J.J.V. and M.A.S. performed experiments and analysed data; C.P., A.A.N., N.D., C.H.S., J.N.B. and M.A.S. wrote the paper; R.C.T., C.H.S., J.J.B. and M.A.S. supervised the project; A.D.B., A.L., V.B., T.D.S. and the Acne Genetic Study Group contributed clinical information and samples; M.A., A.P. and J.A.M. gave technical support.

## Additional information

## The Acne Genetic Study Group

Anton Alexandroff[10], Alex Anstey[11], Jaskiran Azad[12], Omar Aziz[13], Nigel Burrows[14], Aamir Butt[15], Peter Cartwright[16], Anna Chapman[17], Timothy H. Clayton[18], Sandeep Cliff[19], Tim Cutler[13], Brigid Daly[20], Amrit Darvay[21], Claudia DeGiovanni[22], Anthony Downs[23], Colm Dwyer[24], John English[25], Adam Ferguson[26], Colin Fleming[27], Elizabeth Fraser-Andrews[28], Mark Goodfield[29], Clive E. Grattan[30], Hartmut Hempel[31], Sue Hood[13], Bronwyn Hughes[32], Evmorfia Ladoyanni[33], Calum Lyon[34], Ali Mahmud[35], Moshin Malik[36], Eleanor Mallon[37], Simon Meggitt[38], Andrew Messenger[39], Yaaseen Moosa[40], Stephanie Munn[41],

Anthony Ormerod[42], Deepak Rallan[13], Janet Ross[43], Ingrid Salvary[44], Rachel Wachsmuth[45], Shyamal Wahie[46], Shernaz Walton[47], Sarah Walsh[48], Diane Williamson[49] & Carolyn Willis[50]

[10]Leicester Royal Infirmary, University Hospitals of Leicester NHS Trust, Leicester LE1 5WW, UK. [11]St Woolos Hospital, Aneurin Bevan Health Board, Gwent NP20 4SZ, UK. [12]James Cook University Hospital, South Tees Hospitals NHS Foundation, South Tees TS4 3BW, UK. [13]Ipswich Hospital NHS Trust, Ipswich IP4 5PD, UK. [14]Addenbrooke's Hospital, Cambridge University Hospitals NHS Foundation Trust, Cambridge CB2 0QQ, UK. [15]Scunthorpe General Hospital, Northern Lincolnshire and Goole Hospitals NHS Foundation Trust, Northern Lincolnshire DN15 7BH, UK. [16]Queen's Hospital, Burton Hospitals NHS Foundation Trust, Burton DE13 0RB, UK. [17]Queen Elizabeth Hospital Woolwich, Lewisham and Greenwich NHS Trust, Woolwich, London SE18 4QH, UK. [18]Salford Royal NHS Foundation Trust and Royal Manchester Children's Hospital, Manchester M6 8HD, UK. [19]East Surrey Hospital, Surrey and Sussex Healthcare NHS Trust, Redhill RH1 5RH, UK. [20]Royal Blackburn Hospital, East Lancashire NHS Trust, Blackburn BB2 3HH, UK. [21]Southmead Hospital, North Bristol NHS Trust, Bristol BS10 5NB, UK. [22]Brighton General Hospital, Brighton and Sussex University Hospitals NHS Trust, Brighton BN2 3EW, UK. [23]Royal Devon & Exeter Hospital, Royal Devon & Exeter NHS Foundation Trust, Exeter EX2 5DW, UK. [24]Crosshouse Hospital, NHS Ayrshire and Arran, Ayrshire & Arran KA2 0BE, UK. [25]Department of Dermatology, Queen's Medical Centre, Nottingham University Hospitals NHS Trust, Nottingham NG7 2UH, UK. [26]Royal Derby Hospitals, Derby Hospitals NHS Foundation Trust, Derby DE22 3NE, UK. [27]Ninewells Hospital and Medical School, NHS Tayside, Dundee, Dundee DD1 9SY, UK. [28]Essex County Hospital, Colchester CO3 3NB, UK. [29]Chapel Allerton Hospital, The Leeds Teaching Hospitals NHS Trust, Leeds LS7 4SA, UK. [30]Norfolk and Norwich University Hospitals NHS Foundation Trust, Norwich NR4 7UY, UK. [31]Great Western Hospitals NHS Foundation Trust, Swindon SN3 6BB, UK. [32]St Mary's Hospital, Portsmouth Hospitals NHS Trust, Portsmouth PO3 6DW, UK. [33]Corbett Hospital, The Dudley Group of Hospitals NHS Foundation Trust, Dudley DY8 4JB, UK. [34]York Teaching hospitals NHS Foundation Trust, York YO31 8HE North Yorkshire, UK. [35]Eastbourne District General Hospital, East Sussex Healthcare NHS Trust, Eastbourne BN21 2UD, UK. [36]Broomfield Hospital, Mid Essex Hospital Services NHS Trust, Chelmsford, Chelmsford CM1 7ET, UK. [37]Mayday Hospital, Croydon Health Services NHS Trust, Croydon CR7 7YE, UK. [38]Royal Victoria Infirmary, The Newcastle upon Tyne Hospitals NHS Foundation Trust, Newcastle NE1 4LP, UK. [39]Royal Hallamshire Hospital, Sheffield Teaching Hospitals NHS Foundation Trust, Sheffield S10 2JF, UK. [40]St George's Healthcare NHS Trust, St George's, London SW17 0QT, UK. [41]Department of Dermatology, Orpington Hospital, King's College Hospital NHS Foundation Trust, Orpington BR6 9JU, UK. [42]Department of Applied Medicine, Aberdeen, University of Aberdeen, Aberdeen AB25 2ZD, UK. [43]University Hospital Lewisham, Lewisham and Greenwich NHS Trust, Lewisham, London SE13 6LH, UK. [44]James Paget University Hospital NHS Foundation Trust, Great Yarmouth NR31 6LA, UK. [45]Yeovil District Hospital and Musgrove Park Hospital, Taunton and Somerset NHS Trust, Taunton and Somerset TA1 5DA, UK. [46]University Hospital of North Durham, County Durham and Darlington NHS Foundation Trust, Durham DH1 5TW, UK. [47]Hull and East Yorkshire Hospitals, NHS Trust and Hull York Medical School, Hull HU3 2JZ, UK. [48]Department of Dermatology, King's College Hospital NHS Foundation Trust, London SE5 9RS, UK. [49]Glan Clwyd Hospital, Betsi Cadwaladr University Health Board, North Wales LL18 5UJ, UK. [50]Department of Dermatology, Amersham Hospital, Buckinghamshire Healthcare NHS Trust, Amersham HP7 0JD, UK

