## [Peer Review File · Nature Communications]

Reviewer #1 (Remarks to the Author):

Petridis et al. performed a genome-wide association study of acne using 3,823 cases and 16,144 controls and then performed a meta-analysis with summary statistics from a previous study, yielding a total sample size of 26,722. This led to the identification of 20 independent association signals at 15 risk loci. Twelve of these 15 risk loci are novel and include WNT10A, a TF binding site for P63 upstream from SEMA4B and LAMC2. Previous GWAS for acne have revealed three loci in Europeans and two in Han Chinese, although the latter were not replicated in the current study highlighting trans-ethnic differences. Overall this study provides important insights into the genetic basis of acne, and intriguing observations about the role of TGFbeta signalling in the phenotype.

This is a straightforward study with a large cohort of cases and controls. Results of the GWAS scan/meta analysis are believable.

Comments are below:

The authors discuss a gender bias for WNT10A. Could they discuss for the remaining loci?

For SEMA4B the authors state that “Whilst little is known about the biological function of SEMA4B itself, the variant is located within an experimentally defined TP63 transcription factor binding site (OREG1110236) within a broader region of DNAase hypersensitivity”. Could they be more explicit about how the TF binding site was defined? Before discussing the possible role of TP63 they should also confirm this binding experimentally which would make their entire study far more compelling.

In the case of LAMC2 it is not sufficient to state that two SNPs (one associated, and one an eQTL), co-localize since they might not be in LD. If they are not in LD then the identification of an eQTL for LAMC2 at the second SNP is unlikely to be relevant. Similarly, co-localization of other eQTLs within other regions of association should be treated with far more caution.

OVOL1 has previously been identified as an atopic dermatitis locus. Can the authors discuss the relationship between the two associations/phenotypes?

Is there any evidence for genetic interaction between the acne loci?

Minor comments:

Table 1. Provide hg browser version for SNP position.

Reviewer #2 (Remarks to the Author):

Here Petridis et al report on the results of an expanded acne GWAS in a European population. The report advances the field by expanding the number of independent acne associations to 20 (at 15 loci), and identifying some of the putative biologic pathways involved. The methodology of the manuscript is statistically sound and overall, the authors do a good job exploring and interpreting the results. Two minor points:

1) Please specify whether a diagnosis of acne was excluded in the control populations (ELSA and USP), given the high prevalence of acne in the general population. If not excluded, please explicitly state.

2) The biological claim regarding LAMC2 seems overreaching: "We also observe that risk alleles at the 1q25 locus are associated with increased expression of LAMC2, indicating that increased risk of severe acne is the phenotypic consequence at the opposing end of an allelic series of LAMC2 for which biallelic loss-of-function mutations cause the recessive blistering skin disease epidermolysis bullosa." The authors have only shown through public eQTL data that the acne risk variant is associated with increased expression of LAMC2. No experimental data has been provided that LAMC2 overexpression modulates the risk of acne. Given that laminin-5 is a structural basement membrane protein, a biologically plausible explanation for how LAMC2 relates to acne risk is lacking. Therefore, as it is uncertain whether LAMC2 actually mediates acne risk or is a bystander, the claim of "phenotypic consequence" should be scaled back.

Reviewer #3 (Remarks to the Author):

Petridis et al. report the results of a GWAS meta-analysis for acne vulgaris that identified 15 genome-wide significant loci. Three of which have been previously reported. Using the data from the

meta-analysis, the authors carried out a series of typical bioinformatic analyses (identification of credible SNP sets, DEPICT analysis, colocalization with eQTLs) to functionally interpret their findings. The study is rigorous and some of the in silico findings are interesting, though there are no mechanistic experiments to further substantiate them. I have the following questions/comments:

1.) From my understanding, the 15 reported loci are the results of the GWAS meta-analysis of an unpublished GWAS data set and the data from Navarini et al. The title however suggest that these are the results of the new GWAS. I suggest revising the title to reflect the nature of the study more accurately.

Moreover, Table 1 only shows the associations findings from the meta-analysis. Please add the results from the individual GWAS (unpublished + Navarini et al. for the lead SNPs). How many genome-wide significant findings were retrieved in the unpublished GWAS of 3,823 cases vs. 16,144 controls? A table listing the top findings of this unpublished GWAS might be of interest to readers, too. Some of the genes in Table 1 are printed in bold. Please explain in the legend.

2.) Acne is frequent in young adolescents, often starting with the onset of puberty but mostly subsides within the third decade of life. Only a small fraction of patients experiences chronification. Moreover, acne is observed more frequently in males compared to females, which suggest an involvement of hormones. Did the authors consider gender or age stratified analyses? The age- and gender distribution of their study cohort should be stated.

3.) The authors claim the identification of 12 genome-wide significant regions in their meta-analysis but do not provide replication of their findings. Additional association data for acne vulgaris from the UK Biobank have been published by the Neale lab (<http://www.nealelab.is/blog/2017/7/19/rapid-gwas-of-thousands-of-phenotypes-for-337000-samples-in-the-uk-biobank>). How do these results compare to the present findings? Do these data support the present findings? Moreover, a combined analysis of the data may identify additional loci and may enable further insights into the biology of acne vulgaris.

4.) How much of the phenotypic variance in acne vulgaris is explained by the reported loci? What is the estimated (SNP)-heritability for acne vulgaris?

5.) In their introduction, the authors note that acne is an inflammatory disease. Their genetic data and the gene-set enrichment however do not seem to point towards inflammation as a major (early) disease process. What is their explanation? Is there any evidence for an shared genetic basis with other inflammatory diseases? Similarly, males are affected more frequently than females, which suggests an involvement of hormonal pathways in the etiology. Is there any evidence for a biological overlap with other hormonal traits?

6.) It is unclear how the TP63 binding site at 15q26.1 was "experimentally defined".

Acne Manuscript - Response to Reviewers

Reviewers' comments:

Reviewer #1 (Remarks to the Author):

Petridis et al. performed a genome-wide association study of acne using 3,823 cases and 16,144 controls and then performed a meta-analysis with summary statistics from a previous study, yielding a total sample size of 26,722. This led to the identification of 20 independent association signals at 15 risk loci. Twelve of these 15 risk loci are novel and include WNT10A, a TF binding site for P63 upstream from SEMA4B and LAMC2. Previous GWAS for acne have revealed three loci in Europeans and two in Han Chinese, although the latter were not replicated in the current study highlighting trans-ethnic differences. Overall this study provides important insights into the genetic basis of acne, and intriguing observations about the role of TGFbeta signalling in the phenotype.

This is a straightforward study with a large cohort of cases and controls. Results of the GWAS scan/meta analysis are believable.

We thank the reviewer for his remarks and specific comments, where appropriate have amended the manuscript and provided results from additional analysis, which have both clarified and enriched the manuscript.

Comments are below:

The authors discuss a gender bias for WNT10A. Could they discuss for the remaining loci?

We do not observe a sex bias in observed effect sizes at the other 14 loci, we have added a sentence stating this in the main text and a have added in a supplementary figure of a sex stratified forest plot of the lead SNPs at each locus.

For SEMA4B the authors state that "Whilst little is known about the biological function of SEMA4B itself, the variant is located within an experimentally defined TP63 transcription factor binding site (OREG1110236) within a broader region of DNAase hypersensitivity". Could they be more explicit about how the TF binding site was defined? Before discussing the possible role of TP63 they should also confirm this binding experimentally which would make their entire study far more compelling.

The TP63 binding site was experimentally defined in a CHIP-seq experiment performed in human neonatal foreskin keratinocytes, reported in McDade et al Nucleic Acids Res 2012. We have amended the main text to illustrate this experimental validation of the TP63 at this site more clearly and have amended Figure 3 to visualise the read data that define the TP63 binding site from this CHIP-seq experiment.

In the case of LAMC2 it is not sufficient to state that two SNPs (one associated, and one an eQTL), co-localize since they might not be in LD. If they are not in LD then the identification of an eQTL for LAMC2 at the second SNP is unlikely to be relevant. Similarly, co-localization of other eQTLs within other regions of association should be treated with far more caution.

The reviewer raises an important point, but they appear to have misinterpreted our experimental approach to evaluating the co-localisation of association signals (in this specific case the acne susceptibility signal at 1q25.3 and a LAMC2 eQTL). Our approach is not just comparing the marginal

associations at a single SNP or even just stating that two SNPs in the region are associated with acne and LAMC2 expression, which as the reviewer points out is fundamentally flawed. The approach we have taken overcome these concerns, by deploying a regional approach to evaluating co-localisation of association signals, this approach is based upon the observation that if two traits (in this case acne susceptibility and LAMC2 expression) share a causal variant the regression coefficients for either trait against any set of SNPs in the neighbourhood of those variants must be proportional. The approach was originally defined in Plagnol et al Biostatistics 2009 and is implemented in the Coloc software package (<https://cran.r-project.org/web/packages/coloc/index.html>). Whilst this approach was documented in the methods section, we have now amended to main text to explicitly state that we have taken a regional approach to colocalisation.

OVOL1 has previously been identified as an atopic dermatitis locus. Can the authors discuss the relationship between the two associations/phenotypes?

The relationship between the acne and atopic dermatitis association signals near the OVOL1 gene were discussed in detail in the Navarini et al manuscript describing the first acne GWAS in the UK population in which this association at 11q13.1-13.2 was first described, there is no new data in this manuscript that provides new insight into this relationship.

Is there any evidence for genetic interaction between the acne loci?

There is no evidence for genetic interactions between the identified risk loci. We have added a statement to the main text.

Minor comments:

Table 1. Provide hg browser version for SNP position.

We have amended the table to provide this information.

Reviewer #2 (Remarks to the Author):

Here Petridis et al report on the results of an expanded acne GWAS in a European population. The report advances the field by expanding the number of independent acne associations to 20 (at 15 loci), and identifying some of the putative biologic pathways involved. The methodology of the manuscript is statistically sound and overall, the authors do a good job exploring and interpreting the results. Two minor points:

We thank the reviewer for their support of the methodology, relevance of the findings and their presentation in the manuscript.

1) Please specify whether a diagnosis of acne was excluded in the control populations (ELSA and USP), given the high prevalence of acne in the general population. If not excluded, please explicitly state.

The ELSA and USP cohorts are unselected population control cohorts with no recorded information as to a subject's history of acne. We have stated that the cohorts are unselected population cohorts more clearly in the main text and methods section.

2) The biological claim regarding LAMC2 seems overreaching: "We also observe that risk alleles at the 1q25 locus are associated with increased expression of LAMC2, indicating that increased risk of severe acne is the phenotypic consequence at the opposing end of an allelic series of LAMC2 for which biallelic loss-of-function mutations cause the recessive blistering skin disease epidermolysis

bullosa." The authors have only shown through public eQTL data that the acne risk variant is associated with increased expression of LAMC2. No experimental data has been provided that LAMC2 overexpression modulates the risk of acne. Given that laminin-5 is a structural basement membrane protein, a biologically plausible explanation for how LAMC2 relates to acne risk is lacking. Therefore, as it is uncertain whether LAMC2 actually mediates acne risk or is a bystander, the claim of "phenotypic consequence" should be scaled back.

The statement that 'acne risk alleles at the 1q25 locus are associated with increased expression of LAMC2 is valid', but we appreciate the reviewer's comment that remainder of the sentence that infers a direct link between acne risk and LAMC2 expression is over stating the case. We have amended the text to reflect that this is a hypothesis.

Reviewer #3 (Remarks to the Author):

Petridis et al. report the results of a GWAS meta-analysis for acne vulgaris that identified 15 genome-wide significant loci. Three of which have been previously reported. Using the data from the meta-analysis, the authors carried out a series of typical bioinformatic analyses (identification of credible SNP sets, DEPICT analysis, colocalization with eQTLs) to functionally interpret their findings. The study is rigorous and some of the in silico findings are interesting, though there are no mechanistic experiments to further substantiate them. I have the following questions/comments:

We thank the reviewer for their insightful review of our manuscript, where appropriate we have undertaken additional analysis and amended the manuscript.

1.) From my understanding, the 15 reported loci are the results of the GWAS meta-analysis of an unpublished GWAS data set and the data from Navarini et al. The title however suggest that these are the results of the new GWAS. I suggest revising the title to reflect the nature of the study more accurately.

The reviewer raises an important point and we have amended the title of the manuscript to better reflect the study design 'Genome-wide association study and meta-analysis implicates mediators of hair follicle development and morphogenesis as risk factors for severe acne'.

Moreover, Table 1 only shows the associations findings from the meta-analysis. Please add the results from the individual GWAS (unpublished + Navarini et al. for the lead SNPs). How many genome-wide significant findings were retrieved in the unpublished GWAS of 3,823 cases vs. 16,144 controls? A table listing the top findings of this unpublished GWAS might be of interest to readers, too. Some of the genes in Table 1 are printed in bold. Please explain in the legend.

We have amended Table 1 to include the summary stats from the individual GWAS, we have also removed the bold typeface on the genes which was an error in formatting. Eight loci were genome-wide significant in the unpublished GWAS alone, this should now be clear in Table 1.

2.) Acne is frequent in young adolescents, often starting with the onset of puberty but mostly subsides within the third decade of life. Only a small fraction of patients experiences chronification. Moreover, acne is observed more frequently in males compared to females, which suggest an involvement of hormones. Did the authors consider gender or age stratified analyses? The age- and gender distribution of their study cohort should be stated.

We have performed sex stratified analysis, which revealed the difference in observed effect size at the 2q35 locus. There was no evidence of sex bias at any of the other acne susceptibility loci, nor did we identify any additional sex specific associations elsewhere in the genome. This is now stated in the main text. We have not undertaken any age stratified analysis as we currently have incomplete information relating to age of onset in the disease in our cohorts. We have added details of the proportions of males and females in each of the cohorts.

3.) The authors claim the identification of 12 genome-wide significant regions in their meta-analysis but do not provide replication of their findings. Additional association data for acne vulgaris from the UK Biobank have been published by the Neale lab (<http://www.nealelab.is/blog/2017/7/19/rapid-gwas-of-thousands-of-phenotypes-for-337000-samples-in-the-uk-biobank>). How do these results compare to the present findings? Do these data support the present findings? Moreover, a combined analysis of the data may identify additional loci and may enable further insights into the biology of acne vulgaris.

Each of our reported associations are observed with the same direction and magnitude of effect in the new GWAS and the independent study of acne by Navarini et al published in 2014, which represents a validation of these associations. We have considered utilising the UK Biobank as source of additional cases for further investigation of genetic contributors to severe acne, however the study of acne in UK Biobank is currently severely limited by a series of factors; firstly UK Biobank is a cohort of individuals aged between 40-69 at recruitment and has limited records of previously history of acne. In total, only 401 individuals self report a history of acne (of which only 203 are present in the Neale lab analysis) and only 46 individuals in total have a primary or secondary record of acne extracted from hospital episode statistics. Not only do these low numbers illustrate the under reporting (and potentially misreporting) of this clinical diagnosis it also raises questions as to the statistical power of any association test undertaken for this phenotype in this cohort. Indeed, even if one assumes that the self reported cases are representative of the acne population for an effect as large as we observe at the 2q35 we only have ~5% power to detect a variant of this nature. We are investigating with the UK Biobank team approaches that we can improve the reporting and diagnosis of acne, but at this time we don't think adding in additional, and scientifically limited, analysis from UK Biobank will improve the current manuscript.

4.) How much of the phenotypic variance in acne vulgaris is explained by the reported loci? What is the estimated (SNP)-heritability for acne vulgaris?

As suggested by the reviewer, we have performed a SNP heritability estimation using LD score regression. We observe that the SNP heritability on the liability scale is 22% for severe acne. We have also calculated the portion of this SNP heritability and the phenotypic variance of acne that is attributed to the 15 loci associated with severe acne at genome wide significance. We have amended the main text to include this information.

5.) In their introduction, the authors note that acne is an inflammatory disease. Their genetic data and the gene-set enrichment however do not seem to point towards inflammation as a major (early) disease process. What is their explanation? Is there any evidence for an shared genetic basis with other inflammatory diseases? Similarly, males are affected more frequently than females, which suggests an involvement of hormonal pathways in the etiology. Is there any evidence for a biological overlap with other hormonal traits?

We note that there is limited evidence for involvement of either inflammatory or hormonal genes/pathways in the current results. In light of the reviewers comment we have undertake an evaluation of genetic correlation with 175 other traits and observe genetic correlation with

inflammatory bowel disease suggesting there may be elements of shared genetic aetiology with this immune mediated disease, we have briefly added this statement to the manuscript and provided the results of the genetic correlation in a supplementary table. As stated in the discussion, our hypothesis is that the susceptibility loci that we have identified may primarily operating through variation in the structure and maintenance of the hair follicle, creating a microenvironment that is prone to infection and/or inflammation.

6.) It is unclear how the TP63 binding site at 15q26.1 was "experimentally defined".

The reviewer raises a valid question, which was also raised by Reviewer 1. We have now documented how the TP63 binding at 15q26.1 was defined in a CHIP-seq experiment by McDade et al Nucleic Acids Res 2012. We have amended the main text to illustrate this experimental validation of the TP63 at this site more clearly and have amended Figure 3 to visualise the read data that define the TP63 binding site from this CHIP-seq experiment.

Reviewer #1 (Remarks to the Author):

The authors have responded to my concerns and I have no others.

Reviewer #2 (Remarks to the Author):

The authors have satisfactorily addressed this reviewer's concerns. The manuscript is now improved.

Reviewer #3 (Remarks to the Author):

The authors have addressed all my previous points satisfactorily.